# An Active Bio-Based Food Packaging Material of ZnO@Plant Polyphenols/Cellulose/Polyvinyl Alcohol: DESIGN, Characterization and Application

**DOI:** 10.3390/ijms24021577

**Published:** 2023-01-13

**Authors:** Da Song, Li-Wei Ma, Bo Pang, Ran An, Jing-Heng Nie, Yuan-Ru Guo, Shujun Li

**Affiliations:** Key Laboratory of Bio-Based Material Science & Technology (Ministry of Education), College of Material Science and Engineering, Northeast Forestry University, Harbin 150040, China

**Keywords:** plant polyphenol, cellulose, PVA film, zinc oxide, UV shielding, antibacterial activity

## Abstract

Active packaging materials protect food from deterioration and extend its shelf life. In the quest to design intriguing packaging materials, biocomposite ZnO/plant polyphenols/cellulose/polyvinyl alcohol (ZnPCP) was prepared via simple hydrothermal and casting methods. The structure and morphology of the composite were fully analyzed using XRD, FTIR, SEM and XPS. The ZnO particles, plant polyphenols (PPL) and cellulose were found to be dispersed in PVA. All of these components share their unique functions with the composite’s properties. This study shows that PPL in the composite not only improves the ZnO dispersivity in PVA as a crosslinker, but also enhances the water barrier of PVA. The ZnO, PPL and cellulose work together, enabling the biocomposite to perform as a good food packaging material with only a 1% dosage of the three components in PVA. The light shielding investigation showed that ZnPCP−10 can block almost 100% of both UV and visible light. The antibacterial activities were evaluated by Gram-negative *Escherichia coli* (*E. coli*) and Gram-positive *staphylococcus aureus* (*S. aureus*), with 4.4 and 6.3 mm inhibition zones, respectively, being achieved by ZnPCP−10. The enhanced performance and easy degradation enables the biocomposite ZnPCP to be a prospect material in the packaging industry.

## 1. Introduction

The diversity and development of preserving and processing techniques have significantly improved food safety in recent years [1,2,3,4]. Accordingly, active food packaging materials are of tremendous importance in preventing chemical, physical and biological changes during food storage and transportation [5,6]. Several materials have been trialed as active food packaging materials to meet these requirements; however, most of them are either expensive or non-degradable. Thus, searching for new materials that can reduce food-borne diseases and eradicate environmental impacts is necessary [7,8]. In this regard, antibacterial packaging material is a good candidate. It can effectively kill/inhibit the growth of bacteria, prevent food contamination and achieve the long shelf life of food [9,10]. However, how to access an effective packaging material that both works as an antibacterial agent and is useful for human health is still challenging [11,12,13].

Bio-based materials have received increasing attention due to their environmental friendliness and structural diversity [14,15,16,17,18]. Plant polyphenols (PPL) are an abundant natural material and exist in almost all plants. These materials can also work as a natural antioxidant and possess antibacterial activity at the same time [19,20]. The main component of PPL is proanthocyanidins, which are oligomers or polymers of flavan-3-ol units. PPL contains more than 10,000 compounds, each of which has at least one aromatic ring connected by hydroxyl groups [21]. This structural characteristic endows PPL with exciting properties, such as oxidation resistance [22,23], antibacterial activity [24,25], anti-ultraviolet [26] and anti-cancer properties [27,28]. Due to its good physiological activity and its benefits to human health, PPL has been applied as an active food packaging material. However, the addition of small doses of PPL into packaging materials weakens its performance, thus limiting its further application.

Numerous methods have been developed to improve the properties of PPL, for instance, by combining it with other functional materials such as chitosan [29,30,31], anthocyanins [32,33] and tannic acid [34,35]. As an environmentally friendly, non-toxic and wide band gap semiconductor material, Zinc oxide (ZnO) could also be utilized to make a composite with PPL [36,37]. Under light illumination, electrons from ZnO could be excited into the conduction band [38], leaving holes on the valence band. These photo-generated electrons can reduce the adsorbed oxygen on the surface of ZnO and produce oxidative O^2−^ ions, which play a key role in killing bacteria and viruses [39]. It has also been found that ZnO with certain exposed crystal facets has enhanced light sensitivity. Therefore, ZnO is often used as an inorganic UV-shielding and antibacterial agent [40]. However, it also has some common weaknesses, such as its easy agglomeration and tendency to dissolve in acidic conditions, which restricts its further application [41]. Therefore, it is highly desirable to develop a technique that can sufficiently disperse and stabilize ZnO onto PPL [41]. Polyvinyl alcohol (PVA) is low cost and has good film-forming properties, which can be consumed by bacteria and enzymes [14]. Due to hydroxyl groups and hydrogen bonds in its structure, PVA is water-soluble and can be processed by non-petroleum routes. Meanwhile, the film made with PVA is flexible, non-toxic and biocompatible, and it has good mechanical properties. However, the lack of certain properties limits its further application as an active food packaging material.

Using PPL and PVA to build the cross-linked structure along with the functional materials would endow the composite film with an active property. In our previous study, we found that ZnO prepared with the assistance of cellulose preferred growing along (001) crystal facet, and small particles were obtainable [42,43]. Meanwhile, the PPL could improve the combination of ZnO with other polymers via its cross-linkage network. Based on that, a quarternary composite as a biodegradable food packaging material of ZnO@PPL/cellulose/polyvinyl alcohol film (ZnPCP) was designed in this work. The roles of each component of the composite film were studied, and the synergistic effects of these components are addressed. Due to these synergistic effects, the light shielding and antibacterial activity of the composite showed good performance. Due to its being low cost and non-toxic, as well as having good biodegradable properties, ZnPCP offers a good prospect for use as a food packaging material in the future.

## 2. Results and Discussion

The crystalline structures of various ZnPCP samples were studied using an XRD analysis. All the samples showed the similar general patterns seen in Figure 1a. The diffraction peak located at 2θ of 20.0° is the characteristic peak of polyvinyl alcohol (PVA), and the peak appearing at 22.7° belongs to the (002) diffraction plane of cellulose [44]. The diffraction peaks at 31.8°, 34.4° and 36. 3° in the patterns are attributed to the (100), (002) and (101) planes of hexagonal wurtzite ZnO (PDF#36−1451), respectively. The sharpness of these peaks indicates that the ZnO component in ZnPCP had high crystallinity. By increasing the doping ratio, the intensities of ZnO diffraction peaks in composite ZnPCP were greatly enhanced. This means that the content of ZnO in the composite increased. Since the relative intensity of (002) plane became smaller compared to that of bare ZnO (PDF#36−1451), it is evident that the growth along the c-axis was suppressed by cellulose. The controlled growth of ZnO by the cellulose would enhance its property due to the enhancement of the high-energy (001) plane of ZnO. Since the PPL in the composite was amorphous, no diffraction peak of PPL was observed.

The infrared spectra were used to characterize the functional groups, and details of the composite analysis are shown in Figure 1b. It can be seen that all the samples showed almost the same spectral character. The peak centered at 3280 cm^−1^ is assigned to the stretching vibration of hydroxyl group. Due to the diverse environment of hydroxyl groups appearing in PVA, cellulose and PPL, this peak was intense and broad. The C−H stretching vibration could be observed at 2910 cm^−1^, and the characteristic absorption peaks between 1720, 1580 and 1400 cm^−1^ are related to the vibrational modes of C=O and benzene skeleton. Compared to PPL, the vibrations of benzene skeleton of ZnPCP had a blue shift by 20 cm^−1^. This may have been caused by a hydrogen-bonding interaction between PVA and PPL, which weakens the vibrations of the benzene skeleton. The adsorption peaks around 1090 cm^−1^ are attributed to the vibrations of C−O in PVA, cellulose and PPL.

To fully study the interaction between PPL and PVA, relativistic DFT calculations were carried out. Here, we chose three types of PPL basic units, labelled as a, b and c. Based on the finite cluster model, all possible isomeric structures were optimized. The most stable one among each type is illustrated in Figure 1c. The calculated free energy of forming isomers ranged from −0.14 to −0.02 eV, which indicates that the reaction was exothermic and spontaneous. Associated with optimized geometrical results, hydrogen bonds are attributed to the interfacial interaction between PPL and PVA in the composite [45,46,47]. For example, the short interfacial H_PPL_··O_PVA_ distances were calculated to be 1.613, 1.611 and 1.758 Å for the three isomers of PPL/PVA, which have bond orders of 0.22~0.16; moreover, the related O_PPL_−H_PPL_··O_PVA_ angles were close to 180°. The energy decomposition analysis (EDA) showed that PPL−a/PVA had a total bonding energy of −0.63 eV. The other two isomers had interfacial interaction strength in the same magnitude. In brief, PPL and PVA had interfacial chemical coupling in their composite, which was of the hydrogen-bond nature.

SEM images were taken to further explore the morphology of ZnPCP−10. From Figure 2, one can see that both cellulose and ZnO were dispersed in the PVA film. The particle size of covered ZnO was about 1 μm. The cross-section SEM image of ZnPCP film shows that the cellulose on the PVA was 1 μm wide, which would enhance the mechanical strength of the film. The EDS images in Figure 2d provide evidence for the existence of C, O and Zn elements in the composite, while the ZnO content was calculated to be about 8.6%.

The XPS spectra of ZnPCP−10 are shown in Figure 3. The full spectrum (Figure 3a) shows that the film was composed of C, O and Zn elements, which is consistent with the EDS result. Figure 3b is the C1s spectrum, in which three peaks for C−C, C−O and C=O could be fitted. Their corresponding binding energies were 284.8 eV, 286.0 eV and 288.5 eV, respectively. To the O1s spectrum of Figure 3c, three peaks at the binding energies of 531.0, 532.3 and 533.1 eV were also obtained after fitting. The first one is attributed to the lattice oxygen of ZnO, and the last two correspond to C−O−H and C−O−C bonds originating from cellulose, PPL and PVA [16,48,49,50,51]. The Zn2p spectrum showed a typical doublet peak at 1021.9 and 1045.0 eV, which were Zn2p 3/2 and Zn2p 1/2 of divalent Zn element, respectively. Since an XPS analysis can only detect the signals on the surface, the apparent and clear signal of Zn also implied that the coating layer on ZnO was very thin. According to a previous study [43], ZnO is generally coated by PPL. It is deduced that PPL can cross-link with PVA and coats ZnO in the composite. This effect would greatly stabilize ZnO particles when packing different foods during application. The relative mass percentage of ZnO in the composite was 9.7%, which is close to the data obtained by EDS. It is also evident that almost all the ZnO particles in the composite resided near the surface.

The mechanical properties of the ZnPCP film were also tested in terms of tensile strength, elongation at break and Young’s modulus, and the results are shown in Table 1. The tensile strength of the ZnPCP film fell between 30 and 33 MPa, which is about 20 Mpa lower than PVA. This is due to the fact that introducing ZnO into PVA weakens the cross-linkage effect of PVA, and thus, the tensile strength decreases. Moreover, a slightly higher tensile strength of ZnPCP−10 can be obtained with the addition of more ZnO/PPL/cellulose into PVA. This is because the cellulose and PPL in PVA can enhance the linkage by intermolecular/intramolecular hydrogen bonds to some degree [43]. Compared to PVA, the elongation at break decreases and Young’s modulus increases. This indicates that the biocomposite film is difficult to deform [52]. Meanwhile, it can widen the application of ZnPCP as a food packaging material.

It is well known that light irradiation can affect the shelf life of food. Herein, the light shielding effect of ZnPCP films was studied using UV-visible light. Firstly, it can be seen from Figure 4a that the transmittance of PVA film in the UV (200−400 nm) region was more than 80%, and this reached90% for the visible light. This indicates that the PVA film itself had little light shielding effect. In contrast, the transmittance of light between 200 and 600 nm was about 0% for PPL, showing that PPL had a good light shielding effect. If PPL is doped into PVA composite film, both the UV and the visible transmittance of ZnPCP films will be significantly reduced. In ZnPCP−2 with only 0.2 wt% dosage of ZnO@PPL/cellulose, only 3% of UV light and 25% of visible light can pass through the film. When 0.8% and 1% dosages of ZnO@PPL/cellulose are added, the ZnPCP−8 and ZnPCP−10 films can shield almost 100% of both UV light and visible light.

The presence of bacteria is another important factor that causes food spoilage. To test the antibacterial property of the composite films, the inhibition zone method was applied. Typical bacteria of *E. coli* and *S. aureus* were chosen. It can be seen from Figure 4b,c that the PVA film had almost no antibacterial effect on these two bacteria. When ZnO/PPL/cellulose was introduced into PVA, the resultant ZnPCP films showed good antibacterial activity. The increase in the content of ZnO/Cel/PVA in ZnPCP could enhance the antibacterial effect against both *E. coli* and *S. aureus*. Among all the composite samples, the ZnPCP−10 film had the best antibacterial performance, with inhibition zones of 4.4 and 6.3 mm against *E. coli* and *S. aureus*, respectively. This good property is caused by the synergistic effect of cellulose, PPL and ZnO. First, PPL itself has many phenol groups, which can inhibit the growth of bacteria; second, ZnO can produce reactive oxygen species (ROS), which can kill bacteria [39,53,54]. Since the microstructures of ZnO are controlled by cellulose with the exposed high-energy (001) plane, this would enhance the antibacterial property. Compared with the previous study of ZnO@PPL/PVA film, in this study, the antibacterial property of ZnPCP−10 film was improved.

The water content, water solubility and water vapor permeability of the composite film were also studied. The water content of the ZnPCP films was about 9.4%, which is close to that of PVA (9.86%). This indicates that introducing ZnO@PPL/cellulose into PVA has a slight effect on the water content of the film. Figure 5b shows the water solubility of the ZnPCP films. It can be seen that the ZnPCP films had a solubility of about 8.9~6.6%, which is lower than that of PVA. At the same time, with an increased dosage of ZnO@PPL/cellulose, ZnPCP film with a lower water solubility could be obtained. Therefore, it can be speculated that the decrease in water solubility may be caused by interactions between PPL, cellulose and PVA, which would enhance the cross-linkage of PVA by building hydrogen bonding between them. Therefore, ZnPCP film has a better water resistance performance.

One of the important roles of food packaging is to reduce the water exchange between packaged food and the surrounding environment [55]. To evaluate this property, the water vapor permeability (WVP) test was used to study water transfer between the films (Figure 5c). WVP for PVA was tested to be 12.7 g m^−2^ h^−1^ and for ZnPCP films to be between 10.8 and 7.5 g m^−2^ h^−1^. ZnPCP films exhibited a better water vapor barrier property than PVA. The low WVP would prevent water transfer between food and the surrounding environment [56].

Since free radicals in food are related to food deterioration and quality loss, the antioxidant property of packaging materials can also impact food shelf life [57]. Thus, the antioxidant activity of the samples was evaluated by scavenging 2,2-diphenyl-1-propionylhydrazine (DPPH) free radicals [58]. From Figure 5d, one can see that a high content of ZnO@PPL/cellulose in film results in a high oxidation resistance of ZnPCP film. The radical scavenging activity of ZnPCP−10 reached 40.2%. The results show that ZnPCP film has a good application prospect as a packaging material for extending the shelf life of food.

In order to study their fresh-keeping performance, the prepared films were used to pack fresh apple slices, as shown in Figure 6a. The images of the apple slices and their quality change are recorded in Figure 6b,c. Figure 6b shows that the weight loss of the apple slice in a ZnPCP bag was lower than those without a bag or in a PVA bag. Figure 6c contains images of the bagged apple slices at different time intervals. It can be seen that apple slices exposed to air lost water and showed a browning phenomenon in 24 h. The apple slice bagged with PVA film also showed a color change in 24 h, but no water loss could be observed. However, an obvious brown color could be observed in 48 h. When bagged in ZnPCP film for 72 h, only a slight browning phenomenon was observed. This result shows that ZnPCP film has excellent food-preservation performance.

## 3. Materials and Methods

### 3.1. Chemical Reagents

Zn(Ac)_2_·2H_2_O, NaOH, anhydrous ethanol and polyvinyl alcohol (PVA, DP = 1700, 99% hydrolysis, flocculant) were purchased from Tianjin Continental Chemical Reagent Factory. The cellulose was bought from Hangzhou Fuyang Beimu Pulp and Paper Co., Ltd. Agar and beef extract were purchased from Fujian Gaoxin Agar Food Co., Ltd. Peptone was purchased from Beijing Aoxin Biotechnology Co., Ltd. Larch bark was collected from Yakeshi, Inner Mongolia, China.

### 3.2. Extraction of PPL

PPL was extracted from larch bark powder by an ethanol extraction method. In a typical experiment, 20 g of larch bark powder is added into 200 mL of 50% ethanol solution (1 g:10 mL) and shaken at 50 °C for 6 h. In our experiment, after filtration, brown PPL solution was obtained with a concentration of about 1.8 kg L^−1^.

### 3.3. Preparation of the ZnO@PPL/Cellulose Composites

Zn^2+^ solution was obtained by dissolving 3.0 g of Zn(Ac)_2_.2H_2_O into 50 mL of distilled water. Then, 1.0 g of cellulose and 7 mL of PPL solution were added to this Zn^2+^ solution. The pH of the mixture was adjusted to 11 by adding 2 mol L^−1^ of NaOH. After stirring for 0.5 h, the mixture was transferred into a hydrothermal reactor and heated at 100 °C for 10 h. After reaction, the ZnO@PPL/cellulose was obtained by filtering, washing and drying.

### 3.4. Preparation of the ZnPCP Films

ZnPCP films were prepared by a solution casting method [59]. First, 2.6 g of PVA was added into 50 mL of distilled water and heated to 90 °C for 1 h to obtain the PVA solution. After cooling the PVA solution to 45 °C, the ZnO@PPL/cellulose composite in 20 mL of water was added into the PVA solution under continuous stirring. After continuous stirring for 30 min, the ZnPCP casting solution was obtained. Then, 10 mL of casting solution was moved into a culture dish with a diameter of 9 cm and dried, and ultimately, ZnPCP film was formed.

The ZnO@PPL/cellulose dosage varied from 0.2 wt%, 0.5 wt%, 0.8 wt% to 1.0 wt%, and the resultant ZnPCP samples were named ZnPCP−2, ZnPCP−5, ZnPCP−8 and ZnPCP−10, respectively.

### 3.5. Characterizations

An X-ray diffractometer (D/max RB, Tokyo, Japan) was applied to analyze the crystal structure of the samples. A Fourier transform infrared spectrometer (FT−IR, PerkinElmer, Hopkinton, MA, USA) and scanning electron microscopy (JEOL, Tokyo, Japan, JSM−7500F) were used to characterize the functional groups and morphologies of the samples. The chemical environments of the samples were recognized using X-ray photoelectron spectroscopy (XPS, Thermo Fisher Scientific Co., Ltd., Waltham, MA, USA). The UV-visible transmittance of the films was measured by a TU−1950 UV-visible spectrophotometer.

### 3.6. Water Content Test

The ZnPCP (2 cm × 2 cm) was placed in a sealed desiccator with 50% relative humidity at 25 °C. After 48 h, sample was taken out and weighed (w). Then sample was placed in a 100 °C oven to dry the film to constant weight and measured again ($w_{0})$. The equation used for the calculation of water content is as follows:(1)$$Water\;content\left(\%\right)=\frac{w-w_{0}}{w_{0}}\times 100\%\;$$

### 3.7. Water Solubility Test

Firstly, the sample (2 cm × 2 cm) was dried at 105 °C for 24 h and weighed ($W_{i}$). The sample was then immersed in 30 mL of deionized water with gentle stirring at room temperature for 24 h. The sample was taken out, dried at 105 °C for 24 h and weighed ($W_{f}$). The water solubility was, thus, calculated according to Equation (2):(2)$$Water\;solubility\left(\%\right)=\frac{W_{i}-W_{f}}{W_{i}}\times 100\%\;$$

### 3.8. Water Vapor Permeability (WVP)

To test the water permeability, the samples were first treated at 23 °C and 50% humidity for 24 h to achieve a balance. After that, they were placed in a weighing bottle with 2.0 g of dried anhydrous calcium chloride. The weighing bottle was moved to a desiccator with saturated potassium sulfate solution, where the recorded temperature and relative humidity were 23 ± 1 °C and 90 ± 5%, respectively. The sample was weighed after every 2 h. WVP was determined by using the following equation:(3)$$\mathrm{WVP}=\frac{\Delta m}{\mathrm{At}}$$
where WVP is the water vapor permeability in g·m^−2^·h^−1^, Δm is the mass difference between two consecutive measurements (g), A is the area of the weighing bottle (m^2^) and t is the time interval between two measurements (h).

### 3.9. Antioxidant Activity

The antioxidant activity of the sample was studied using DPPH. The DPPH solution was prepared by dissolving 1.0 mg of DPPH in 20 mL of ethanol solution (the volume ratio of anhydrous ethanol to deionized water was 4:1). The 0.2 g sample was added to a 5 mL prepared DPPH solution and shaken in an oscillator for 24 h in a dark environment at room temperature [60]. The absorbance of the mixture was measured under 517 nm by a UV spectrophotometer. The scavenging rate was calculated according to the following equation:(4)$$\mathrm{DPPH}\;\mathrm{scavenging}\;\mathrm{activity}\left(\%\right)=\frac{A_{0}-A}{A_{0}}\times 100\%$$
where DPPH scavenging activity represents the free radical scavenging rate of DPPH, $A_{0}$ is the absorbance of DPPH solution at 517 nm and A stands for the absorbance of the mixture.

### 3.10. Antibacterial Experiment and Inhibition Zone Method

To prepare the medium, 3.0 g of beef extract, 10.0 g of peptone, 5.0 g of sodium chloride and 15.0 g of agar were added into 1000 mL of deionized water. The medium was then kept in an incubator at 37 °C for 24 h. Sterilization was performed in a portable pressure steam sterilizer at 121 °C and 0.1 kPa for 20 min. The strains of *Staphylococcus aureus (S. aureus)* and *Escherichia coli (E. coli)* were 1.0 × 10^−4^ CFU mL^−1^. The ZnPCP films were cut into discs with diameters of 15 mm and placed in the culture dish containing solid medium. The culture dish was then placed in an incubator at 37 °C for 24 h. Antibacterial radii were then calculated by using Equation (5):(5)$$A\mathrm{ntibacterial}\;\mathrm{ring}\;\mathrm{radius}=\frac{d_{1}-d_{2}}{2}\left(\mathrm{mm}\right)$$
where d_1_ denotes the diameter of the outer ring of bacteriostatic sphere, and d_2_ is the diameter of the sample (the diameter of the inner ring of the inhibition zone).

### 3.11. Computational Details

The structures of PPL and PVA and their composite isomers were fully optimized using the Priroda code [61]. The scalar all-electron relativistic Hamiltonian, Perdew−Burke−Ernzerhof (PBE) functional and double-zeta polarized basis sets were used. The frequency calculation was used to confirm that the optimized structure was at the minimum point on the potential energy surface. At the same time, thermodynamic data (at 298.15 K) were obtained. To explore the interfacial interaction in the composite, an energy decomposition analysis (EDA) was conducted, which is implemented in the ADF2014 code [62]. The scalar zeroth-order regular approximation (ZORA) Hamiltonian associated with the PBE functional and Slater-type TZP basis sets was employed. The Grimme D3 correction was added to precisely describe the weak dispersion interaction.

## 4. Conclusions

Starting from the raw materials of zinc acetate, PPL, cellulose and PVA, ZnPCP biocomposite film was successfully prepared using the hydrothermal synthesis and solution casting methods. The structural and morphological characterizations show that the growth of ZnO is tuned by cellulose, which leads to the exposure of the high-energy (001) plane. PPL covered the ZnO particles, allowing ZnO to disperse well on the PVA film. Although ZnO was coated by PPL and PVA, the coating layer was thin, as evidenced by the XPS analysis. The special structural feature could benefit ZnO, enabling it to function as an antibacterial and UV shielding agent. It was found that the best sample, ZnPCP−10, blocked almost 100% of UV and visible light. Meanwhile, it also had good antibacterial activity, with inhibition zones of 4.4 and 6.3 mm against *E. coli* and *S. aureus*, respectively. Due to the merits of its low cost, biodegradability and good properties, the newly synthesized ZnPCP could have tremendous potential as a packaging material.

## Figures and Tables

**Figure 1 ijms-24-01577-f001:**
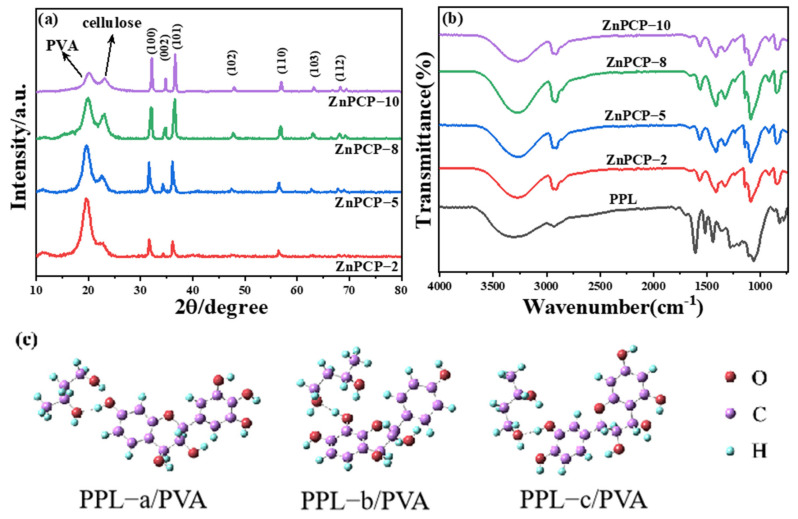
(**a**) XRD patterns of ZnPCP; (**b**) FT−IR spectra of ZnPCP and PPL extract; and (**c**) optimized structures for PPL/PVA isomers.

**Figure 2 ijms-24-01577-f002:**
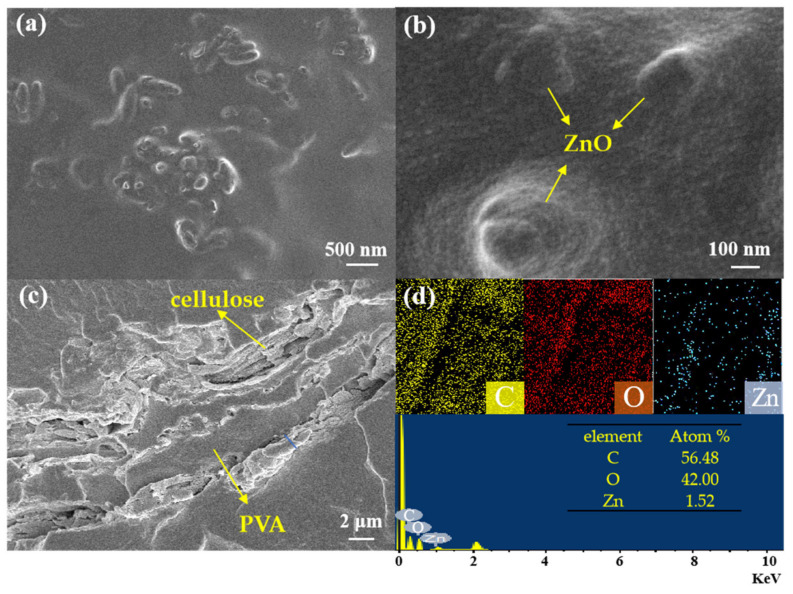
(**a**–**c**) SEM images of ZnPCP and (**d**) EDS mapping images of ZnPCP.

**Figure 3 ijms-24-01577-f003:**
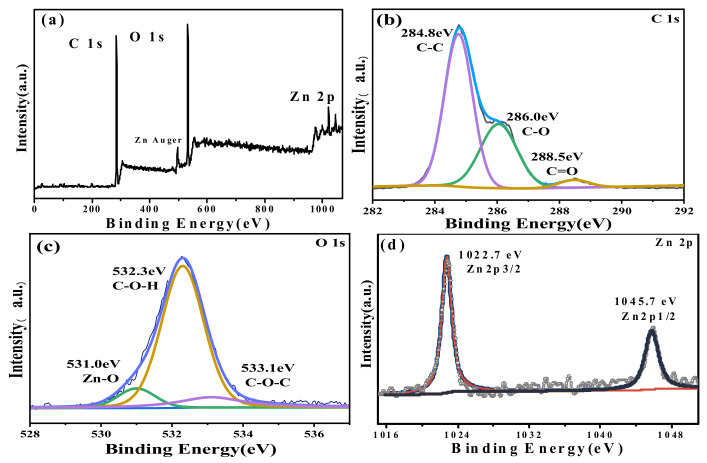
XPS spectra of ZnPCP−10, including (**a**) survey, (**b**) C 1s, (**c**) O 1s and (**d**) Zn 2p.

**Figure 4 ijms-24-01577-f004:**
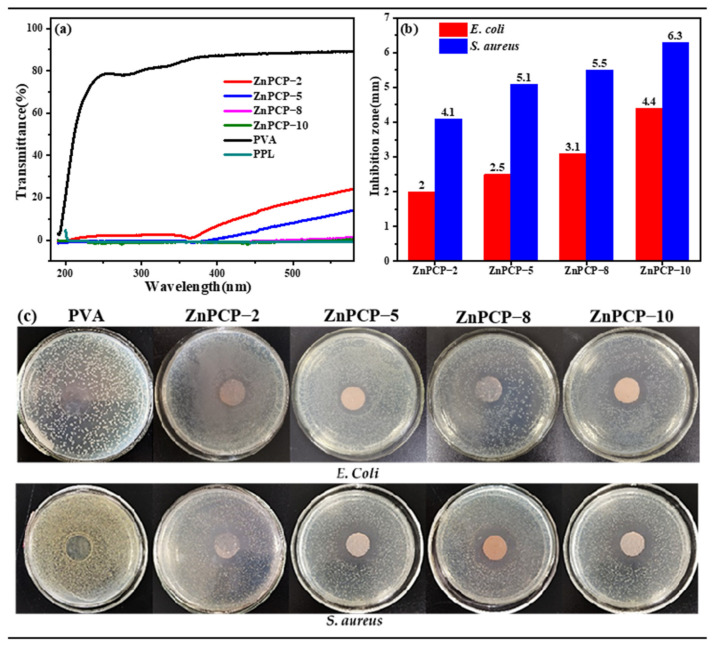
(**a**) UV transmittance of PVA, PPL and ZnPCP films with different mass fractions and (**b**,**c**) antibacterial effect of ZnPCP.

**Figure 5 ijms-24-01577-f005:**
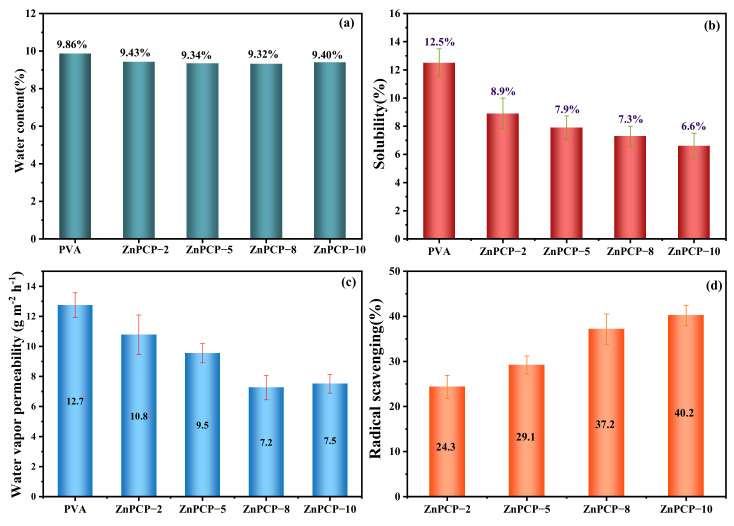
(**a**) The water contents, (**b**) water solubility and (**c**) water vapor permeability of PVA and ZnPCP films, together with (**d**) the antioxidant activity of ZnPCP.

**Figure 6 ijms-24-01577-f006:**
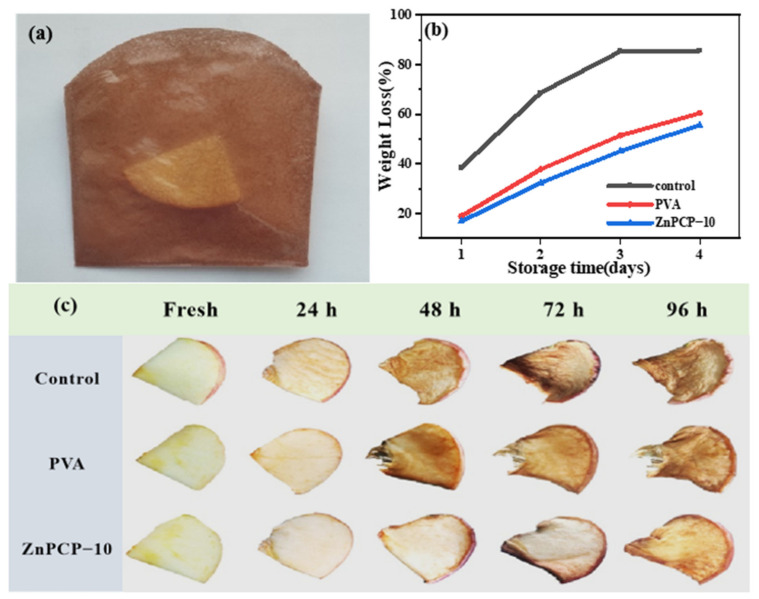
(**a**) ZnPCP−10 bag with apple slice; (**b**) weight loss and (**c**) fresh-keeping performance for bagged apple slices.

**Table 1 ijms-24-01577-t001:** The mechanical behavior of PVA film and ZnPCP.

Sample	Tensile Strength (Mpa)	Elongation at Break (%)	Young’s Modulus (Mpa)
PVA film	56.8 ± 4.9	104.5 ± 16.3	484.3 ± 97.4
ZnPCP−2 film	30.7 ± 1.5	29.5 ± 8.3	549.3 ± 75.1
ZnPCP−5 film	31.3 ± 1.4	19.4 ± 4.5	649.9 ± 101.1
ZnPCP−8 film	32.0 ± 1.8	13.8 ± 0.1	675.9 ± 89.3
ZnPCP−10 film	32.5 ± 0.9	10.7 ± 0.7	702.8 ± 45.3

## Data Availability

Not applicable.

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
