# Peer review of "An Active Bio-Based Food Packaging Material of ZnO@Plant Polyphenols/Cellulose/Polyvinyl Alcohol: DESIGN, Characterization and Application"

_ijms, 2023, doi:10.3390/ijms24021577_

Round 1

Reviewer 1 Report

1-     I wonder whether the solubility of PVA film is only 5%. It’s far from reality.

2-     Why PVA???

3-     What about the interaction between PVA and extracted solution?

4-     Characterization of the extracted materials should be conducted to make sure that the extracted materials were pure polyphenols not containing sugars and other impurities.

5-     Why polyphenols and not anthocyanins?

6-     Please provide a typical image of the resulting films used as apple slice packaging materials rather than only showing the picture of the apple slice.

7-     What about the novelty of this research?

8-     Discussion needs to be improved and justification is needed.

9-     There are a number of related papers that are even not being addressed in this manuscript:

https://doi.org/10.1016/j.carbpol.2021.118550

https://doi.org/10.1016/j.carbpol.2022.119910

I would like to see the required revisions for reconsidering its potential for publication.

Reviewer 2 Report

The paper will be ready for publication after major revision.

Please revise the manuscript according to the attached file.

Round 2

Reviewer 1 Report

Based on the changes made to the manuscript, I believe that this work is suitable for publication.

Author Response

Thanks for the advice for our manuscript. The English language and style of manuscript have been carefully checked and revised.

Reviewer 2 Report

Accept.

Author Response

Thanks for the advice for our manuscript. The English language and style of manuscript has been carefully checked and revised.